# Viral-induced alternative splicing of host genes promotes influenza replication

Matthew G Thompson[1,2], Mark Dittmar[3], Michael J Mallory[2], Prasanna Bhat[4], Max B Ferretti[2,3], Beatriz MA Fontoura[4], Sara Cherry[1,2,3], Kristen W Lynch[1,2]*

[1]Biochemistry and Molecular Biophysics Graduate Group, University of Pennsylvania, Philadelphia, United States; [2]Department of Biochemistry and Biophysics, University of Pennsylvania, Philadelphia, United States; [3]Department of Pathology, Perelman School of Medicine, University of Pennsylvania, Philadelphia, United States; [4]Department of Cell Biology, UT Southwestern Medical Center, Dallas, United States

**Abstract** Viral infection induces the expression of numerous host genes that impact the outcome of infection. Here, we show that infection of human lung epithelial cells with influenza A virus (IAV) also induces a broad program of alternative splicing of host genes. Although these splicing-regulated genes are not enriched for canonical regulators of viral infection, we find that many of these genes do impact replication of IAV. Moreover, in several cases, specific inhibition of the IAV-induced splicing pattern also attenuates viral infection. We further show that approximately a quarter of the IAV-induced splicing events are regulated by hnRNP K, a host protein required for efficient splicing of the IAV M transcript in nuclear speckles. Finally, we find an increase in hnRNP K in nuclear speckles upon IAV infection, which may alter accessibility of hnRNP K for host transcripts thereby leading to a program of host splicing changes that promote IAV replication.

**\*For correspondence:**
klync@pennmedicine.upenn.edu

**Competing interests:** The authors declare that no competing interests exist.

## Introduction

Influenza A virus (IAV) is a ubiquitous and significant health threat, resulting in 290,000–650,000 deaths per year worldwide (**WHO, 2019**). In the United States alone, IAV is estimated to result 12,000–56,000 deaths annually (**Centers for Disease Control and Prevention, 2020**), with an estimated $11.2 billion cost burden (**Putri et al., 2018**). While efforts are ongoing to treat the virus, there still is no universal prevention or cure. This lack of treatment is due, in part, to the virus's ability to rapidly mutate and develop resistance (**Hussain et al., 2017**). Therefore, it is important to further understand how IAV and host cells interact during infection in order to develop new avenues for antiviral therapies.

IAV infection results in large changes to the host gene expression landscape. Our understanding of these changes has largely been shaped through studies using microarray technology, limiting analyses to the measurement of transcript abundance. However, several recent studies have utilized advances in next-generation sequencing to observe host and viral gene expression during infection in greater detail than ever before. For example, IAV infection has been shown to induce a range of gene expression phenotypes such as RNA polymerase readthrough (**Bauer et al., 2018**; **Heinz et al., 2018**; **Zhao et al., 2018**), translational shutoff (**Bercovich-Kinori et al., 2016**), altered genome architecture (**Heinz et al., 2018**), and targeted transcript degradation (**Gaucherand et al., 2019**). In addition, a recent transcriptome-wide study identified not only transcript abundance changes but also global changes in alternative splicing isoforms (**Fabozzi et al., 2018**). These observations are in agreement with a growing awareness that viral infection can not only alter gene expression but also lead to altered splicing patterns.

Host splicing changes have been observed during infection with DNA viruses (HSV-1 [*Hu et al., 2016*], HCMV [*Batra et al., 2016*]) as well as RNA viruses such as reovirus (*Boudreault et al., 2016*), dengue virus (*Sessions et al., 2013*), and zika virus (*Hu et al., 2017*), and recently, SARS-CoV-2 (*Banerjee et al., 2020*). However, these studies represent only the tip of the iceberg as they have mostly lacked the sequencing depth, and subsequent orthogonal validation, that is needed for a comprehensive quantitative analysis of host splicing (*Mehmood et al., 2020*; *Shen et al., 2012*). Therefore, we set out to comprehensively define the scope of host splicing changes during IAV infection in order to begin to address the underlying mechanisms and functional implications of such viral-induced alterations to the host transcriptome.

Mechanistically, changes in host splicing under any condition are generally due to regulation of RNA-binding proteins that, in turn, control the host splicing machinery. Splicing factors bind at a myriad of sites throughout a nascent pre-mRNA, and influence the inclusion or exclusion of specific exons, and in some cases introns, by influencing the recruitment of spliceosomal components to splice sites (*Fu and Ares, 2014*). In some cases, it has been shown that that viruses directly interact with the splicing machinery to alter splice site choice. For example, dengue virus NS5 alters splicing through interaction with the U5 snRNP (*De Maio et al., 2016*) component of the spliceosome, the piconaviral RdRp interacts with RNA helicase Prp8 to stall splicing (*Liu et al., 2014*), and SARS-CoV-2 Nsp16 has been proposed to bind U1 and U2 snRNA (*Banerjee et al., 2020*). Viruses can also influence the localization and/or expression of splicing factors, as has been observed in poliovirus (*Álvarez et al., 2013*), rotavirus (*Dhillon et al., 2018*), alphavirus (*Barnhart et al., 2013*), HCV (*Shwetha et al., 2015*), picornavirus (*Florez et al., 2005*), HCMV (*Batra et al., 2016*), and Dengue virus (*Pozzi et al., 2020*) and the resulting change in the nuclear concentration of splicing regulatory proteins likely contributes to alternative splicing of host genes. In the case of IAV, however, it is unclear how and if viral interactions with the splicing machinery impact host splicing patterns. While it is known that the IAV protein NS1 inhibits splicing in vitro through interaction with the spliceosomal U6 snRNA, the relevance of this interaction in cells has not been shown (*Qiu et al., 1995*).

Here, we demonstrate that infection of human cells with IAV induces widespread alternative splicing changes in host genes. Similar changes in splicing are not observed upon treatment with interferon, demonstrating that this program of IAV-induced splicing is a direct consequence of viral infection rather than a secondary bystander effect of the host innate immune response. Importantly, at least 30 of the 61 genes tested that harbor IAV-induced splicing events influence IAV infection. Directly manipulating splicing of several of these transcripts confirmed that IAV-induced changes in alternative splicing have direct consequences on viral replication. Mechanistically, we found that almost 25% of the IAV-induced splicing changes are also regulated by hnRNP K, a well-known splicing regulator whose subnuclear distribution we show to be altered upon IAV infection. Previously, we demonstrated that hnRNP K promotes splicing of the IAV M segment transcript to generate the M2 protein (*Mor et al., 2016*; *Thompson et al., 2018*; *Tsai et al., 2013*). This suggests that the virus co-opts hnRNP K to promote infection both directly by inducing viral gene splicing and indirectly by altering hnRNP K splicing on pro-viral transcripts altogether promoting replication. Altogether, this work demonstrates the importance of alternative splicing for robust infection and replication of IAV and suggests mechanisms for how such viral-induced regulation of host splicing is achieved.

## Results

### Host gene expression changes are induced upon IAV A/WSN/33 infection

As a first step toward understanding the impact of IAV infection on the transcriptome, human epithelial A549 cells were infected with IAV A/WSN/33 at a multiplicity of infection (moi) of 2, to yield ~90% infection of cells (*Condit, 2013*), and total RNA was isolated at 0, 6, and 12 hr post infection (HPI). These experiments were performed in biological triplicate. mRNA from each replicate was poly(A)-selected and used to make cDNA libraries for 150 nucleotide, paired-end, Illumina Hi-Seq sequencing. Reads were then mapped to human and viral genomes as outlined in *Figure 1a*. At 6 and 12 HPI, IAV RNA accounted for >30% of mapped reads (*Figure 1b*). Despite the large percentage of reads mapping to the viral genome, an average of 60–70 million reads were mapped to the host genome (*Figure 1—figure supplement 1*). Importantly, this depth falls within the

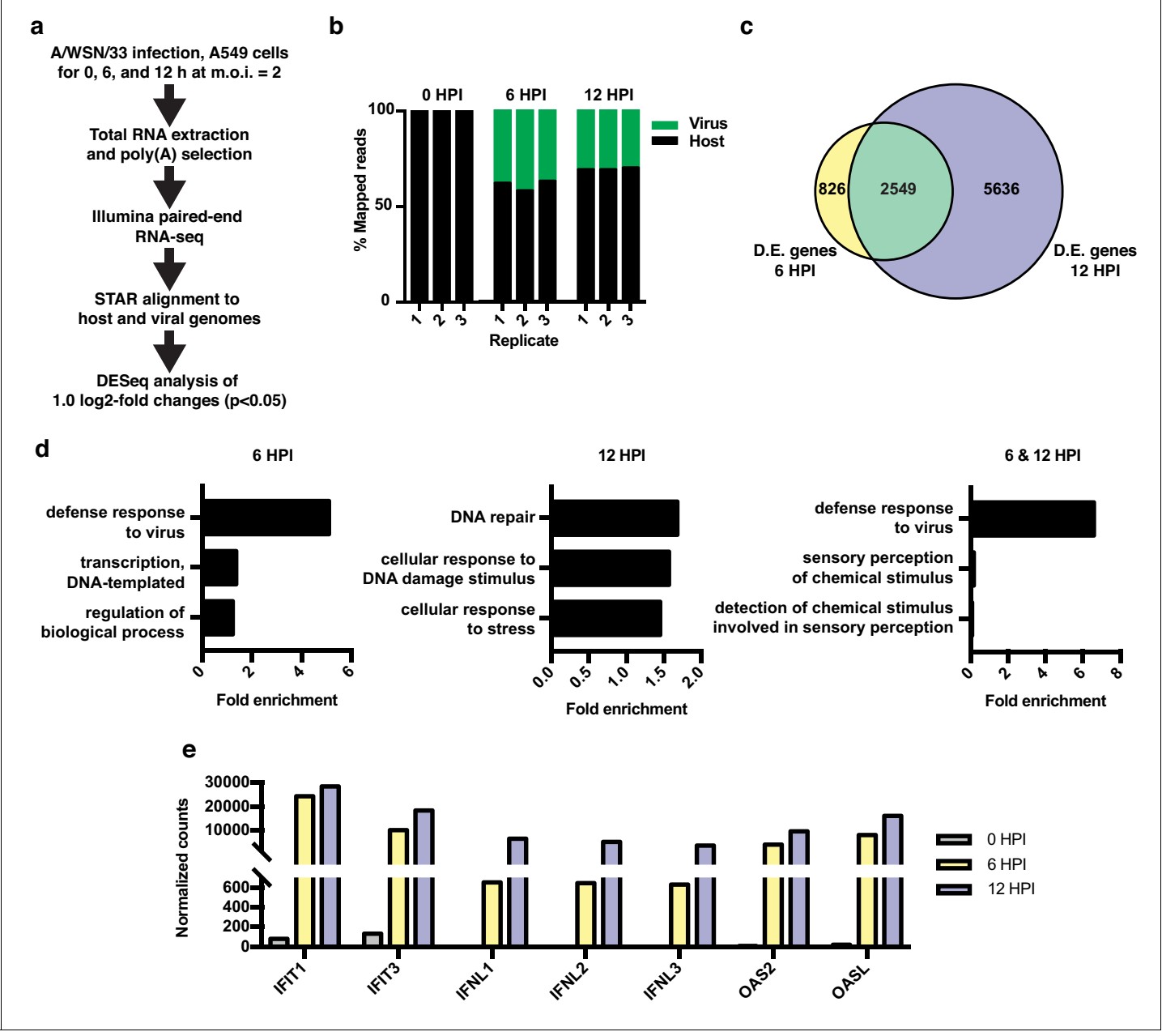

**Figure 1.** RNA-seq and differential expression analyses of IAV infected A549 cells. (**a**) Schematic of infections, RNA-seq, and data processing work-flow. (**b**) Proportion of aligned, mapped reads corresponding to host (hg19) and virus (A/WSN/33) in each of the indicated samples. (**c**) Overlap of differentially expressed genes with a log2 fold change >1.0 at 6 and 12 hr PI. (**d**) Top gene ontology categories for differentially expressed genes enriched at 6 hr PI (left), 12 hr PI (middle), and overlap of 6 and 12 hr PI (right). p-Values were <0.05 in all categories. (**e**) Normalized counts of known viral-responsive genes.

The online version of this article includes the following figure supplement(s) for figure 1:

**Figure supplement 1.** Read mapping to the human and viral genomes.

recommended depth (~60 million reads) for robust and reproducible splicing and differential expression analyses (*Mehmood et al., 2020*; *Shen et al., 2012*).

Differential expression analysis of host transcripts revealed 3375 changes (log2 fold >1, p<0.05) at 6 HPI and 8185 changes at 12 HPI, with an overlap of 2549 changes (*Figure 1c*, list of differentially expressed genes found in *Supplementary file 1*). Gene ontology (GO) analysis of changes at 6 HPI showed an enrichment for genes associated with host response to virus, whereas at 12 HPI, GO

terms were enriched for categories such as DNA repair and stress (*Figure 1d*). GO analysis of the 2549 genes differentially expressed at both 6 and 12 HPI (*Figure 1c*, green overlap) also showed the expected enrichment for host response to virus. We further confirmed the robust induction of individual genes that are frequently used as markers of the innate antiviral immune response by RT-qPCR at both 6 and 12 HPI (*Figure 1e*). Together these data show that under the conditions of our experiment, IAV A/WSN/33 infection induces a robust host innate immune response within 6 hr. By 12 HPI, we observe a further progression of infection and a corresponding stress response in the host.

## Influenza A/WSN/33 infection results in differential splicing of host transcripts

Next, we quantified global changes in host splicing changes using a previously established algorithm, MAJIQ, that identifies all standard splicing events as well as complex splicing patterns (*Vaquero-Garcia et al., 2016*). Importantly, the detection of splicing by MAJIQ is not based on any pre-determined identification of exons, rather MAJIQ can identify any de novo pattern of splicing observed in the RNA-Seq reads (*Vaquero-Garcia et al., 2016*). At 6 HPI, we identified 1162 splicing differences at 861 unique genes with a threshold for changes in percent isoform expression (ΔPSI) of greater than or equal to 10% (*Figure 2a,b*). Using an even more stringent cut-off of 20% ΔPSI, we found 1931 splicing changes at 12 HPI, 896 of which overlapped with those splicing events observed to vary at 6 hr (*Figure 2a,b*; lists of all splicing changes are provided in *Supplementary file 2*). We

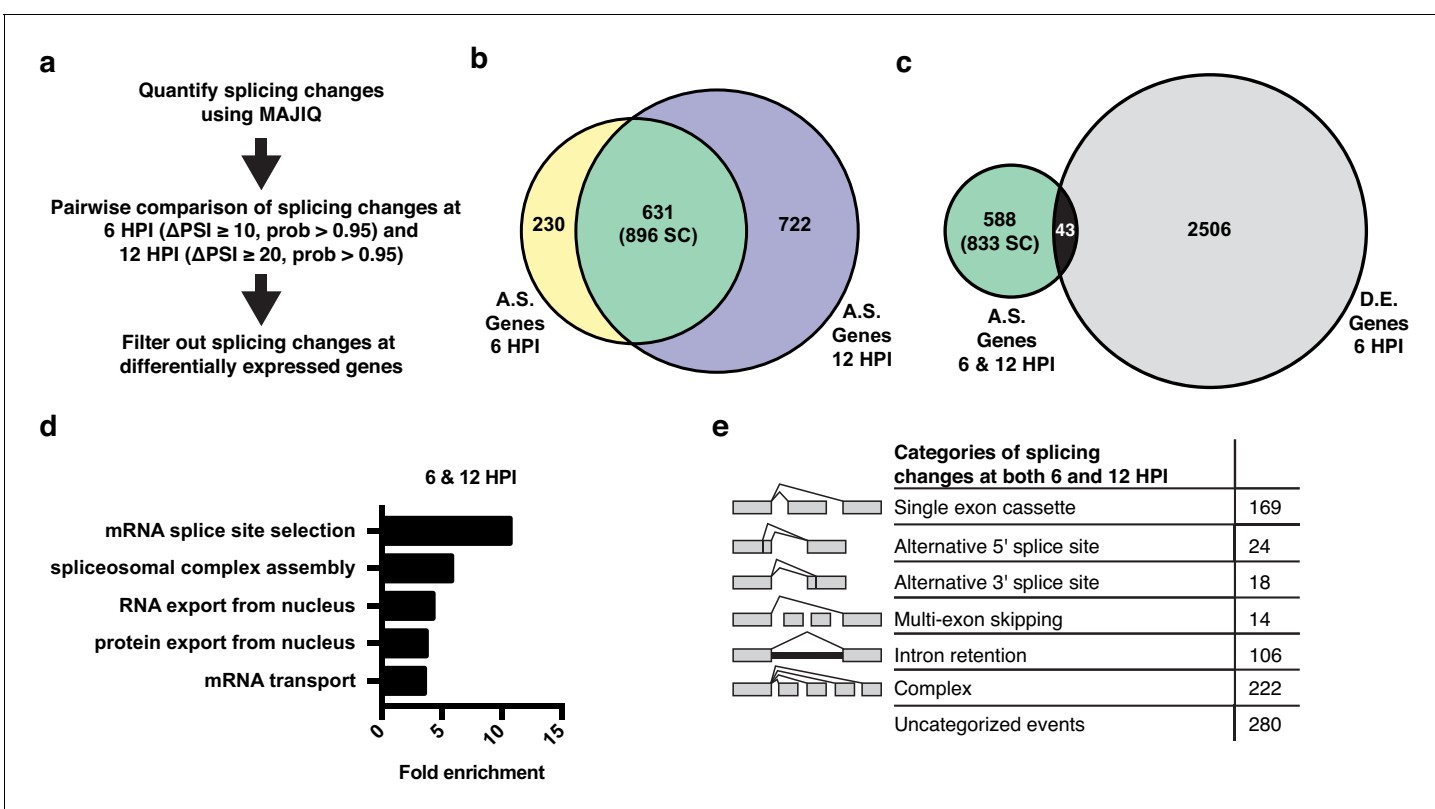

**Figure 2.** Alternative splicing analyses of host genes during IAV infection. (**a**) Schematic of splicing analyses and downstream processing pipeline. (**b**) Overlap of alternatively spliced genes (ΔPSI ≥ 10 at 6 HPI (hours post-infection), ΔPSI ≥ 20 at 12 HPI) at 6 and 12 HPI (number of unique splicing changes (SC) shown in parentheses). (**c**) Overlap of alternatively spliced and differentially expressed genes at 6 and 12 HPI (number of unique splicing changes (SC) shown in parentheses, log2 fold change >1.0). (**d**) Top gene ontology categories enriched within 588 exclusively alternatively spliced genes at 6 and 12 HPI. p-Values were <0.05 in all categories. (**e**) Categorization of 833 exclusively alternative splicing events (588 genes) at 6 and 12 HPI.

The online version of this article includes the following figure supplement(s) for figure 2:

**Figure supplement 1.** Details of splicing and gene expression changes in IAV-infected cells.

used a more stringent threshold at 12 HPI based on the fact that when plotting the ΔPSI values of the overlapping 896 splicing changes against each other, we see a trend in which splicing changes increase during the course of infection (*Figure 2—figure supplement 1*). This trend in increased ΔPSI over time is consistent with host splicing being regulated directly in response to viral load. We have therefore focused our further analysis on the 896 splicing events that are observed at both 6 and 12 hr, as these are both the most reproducible and most correlated with virus levels.

We are particularly interested in virus-induced changes in splicing, rather than those induced as a secondary or cell-extrinsic consequence of innate immune signaling. Therefore, we tested whether the splicing changes induced by IAV infection are also regulated directly by interferon, which is produced by viral-infected cells. Importantly, we found that of 10 splicing events we validated in orthogonal RT-PCR assays to change splicing during viral infection, none exhibit splicing regulation upon treatment with interferon beta (IFNβ), despite the fact that this treatment induces expression of known IFNβ-target genes (*Figure 2—figure supplement 1*). Therefore, we conclude that the majority of the changes in splicing we observed upon IAV infection are a direct consequence of viral load and not a secondary effect of interferon signaling.

The 896 infection-induced splicing events from *Figure 2b* are encompassed in 631 genes. Notably, only 43 of these genes also exhibit changes in abundance upon viral infection (*Figure 2c*), consistent with many other studies that have found that genes regulated by splicing and transcription are largely distinct (*Gazzara et al., 2017*; *Ip et al., 2007*; *Shinde et al., 2017*). Moreover, GO analysis of the 588 genes that are differentially spliced without changes in abundance, finds an enrichment for genes associated with RNA processing, not immune defense (*Figure 2d*). In summary, IAV infection induces a host splicing program that is functionally distinct from the host transcriptional response.

To focus on genes that are regulated solely by splicing and not expression we have removed the 43 differentially spliced and expressed genes from our list of splicing events in the remainder of our analysis, leaving 833 distinct splicing changes in 588 genes. These 833 overlapping splicing events were then categorized by the type of alternative splicing (*Figure 2e*). Of the alternative splicing events, ~25% are 'classic' splicing events falling into alternative 5′ splice site, alternative 3′ splice site, and simple cassette exon events (211 splicing changes total). We also find many events classified as either complex or uncategorized, consistent with previous studies using MAJIQ (*Vaquero-Garcia et al., 2016*). While interesting, these complex events are challenging to explore in terms of functional consequence or mechanism (*Vaquero-Garcia et al., 2016*). Therefore, we focused our subsequent studies on the more canonical splicing events.

## Infection-induced splicing events encode proteins important for IAV replication

We next investigated the role of the genes harboring viral-regulated splicing events in viral infection, focusing on the 211 classic splicing events (single exon cassettes, alternative 5′ splice site, and alternative 3′ splice site) due to their robustness and ease of analysis by RT-PCR (*Figure 2e*). We further filtered this set of 211 events for alternative splicing occurring within the coding sequence of transcripts, which would potentially alter the protein sequence. This resulted in a list of 61 alternatively spliced candidate genes (*Figure 3a*). The functional importance of these 61 genes was tested using an siRNA-based screen to identify if loss of any of these genes alters IAV infection. Briefly, A549 cells were transfected with siRNAs (two independent siRNAs per gene) for 48 hr (*Figure 3—figure supplement 1*), then single-cycle IAV infections were performed for 24 hr followed by scoring of IAV infected cells by staining for IAV NP protein via immunofluorescence. We also quantified cell viability and removed any candidate gene where cell viability was reduced by 25% or more after siRNA treatment. Positive 'hits' were defined by genes where one or both siRNAs induced a greater than 20% change in the number of IAV-positive cells in duplicate screens (*Figure 3*, *Figure 3—figure supplement 1*). Given these parameters, we find that knock down of 25 of the 61 genes yielded a change of greater than 20% in the number of IAV-positive cells (*Figure 3b,c*).

We also wanted to capture genes that might be important for viral particle production and subsequent re-infection rather than initial infection. Thus, for the seven genes that maintained cell viability but did not alter NP expression in our initial screen, we also carried out a multi-cycle infection assay in which we scored viral titer levels. Notably, we found that knockdown of five of these seven genes resulted in dramatic (≥5.5 fold) decreases of infectious viral particles (*Figure 3d*). As a control, we

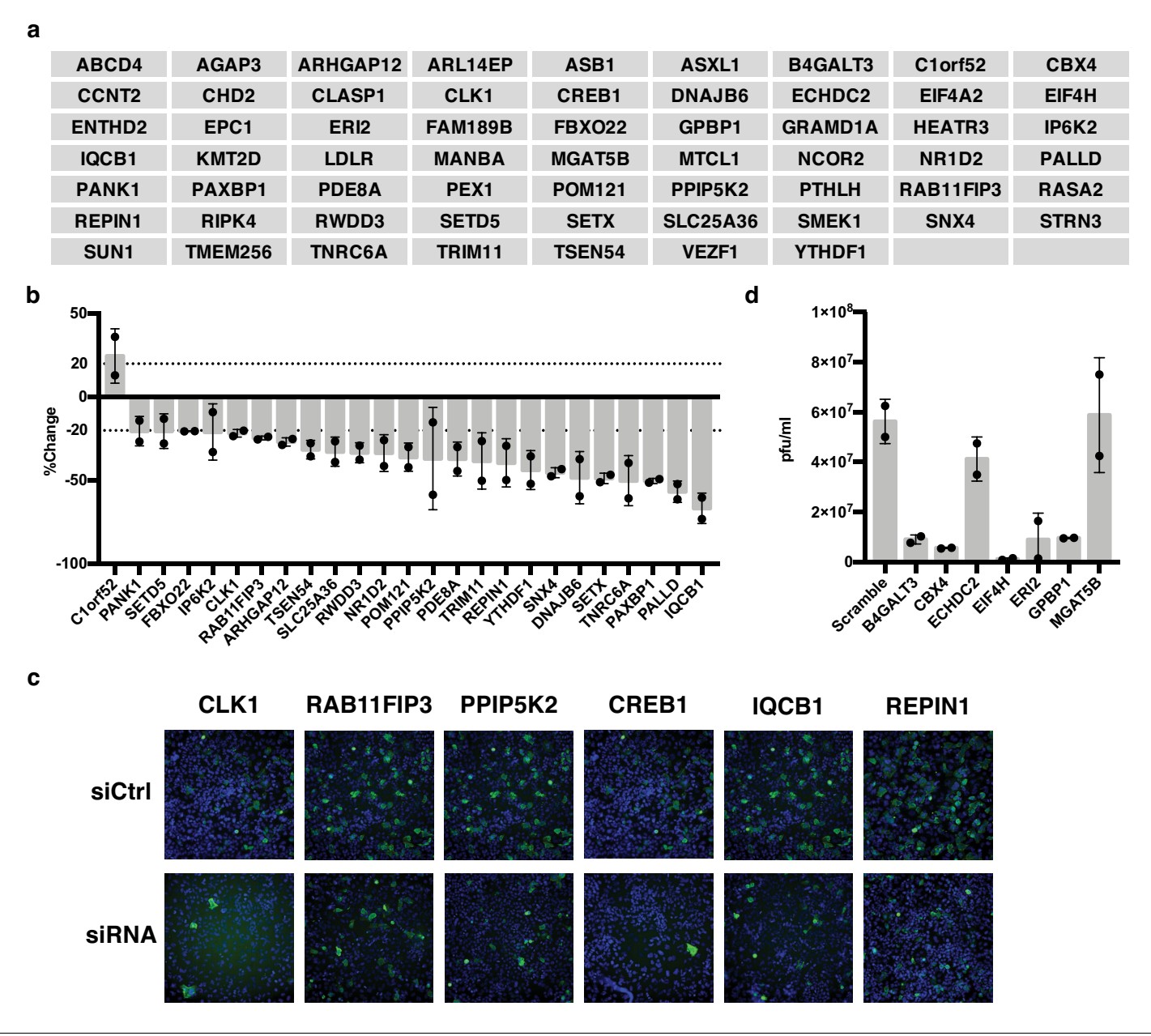

**Figure 3.** IAV-sensitive splicing events encode genes that modulate IAV replication. (**a**) List of 61 alternatively spliced genes sensitive to IAV infection chosen for siRNA screening. (**b**) Percent change of IAV-positive scoring cells in the context of siRNA knockdown. Bar indicates the average of two siRNAs in two independent screens, dots and error bars represent individual screen averages (across two siRNAs) and standard deviation respectively. Genes with an average change of ≥20% are plotted. Quantification of knock-down of representative genes and average results from individual siRNAs are shown in *Figure 3—figure supplement 1b*. (**c**) Representative images of immunofluorescence imaging of IAV-infected cells in the context of control or gene targeting siRNA treatments. Blue staining denotes individual nuclei (DAPI) and green staining denotes IAV-positive cells (NP). (**d**) A549 cells were treated with 20 nM siRNA for 72 hr and infected with IAV at moi = 0.01 for 48 hr. Shown is the measurement of plaque forming units in media via plaque assay. Bar indicates the average of two independent screens, dots and error bars represent individual data points.

The online version of this article includes the following figure supplement(s) for figure 3:

**Figure supplement 1.** Further data and controls for siRNA screen.

tested five genes that showed a difference in infection in the initial screen and all also showed a change in viral titer that is consistent with the screen (*Figure 3—figure supplement 1*) Therefore, taken together, approximately half of the alternatively spliced genes we tested (30/61) influence the IAV infection cycle. We note that while CREB1 has previously been reported to be pro-viral (*König et al., 2010*), we see little impact of siRNAs of this gene in our screen despite both siRNAs decreasing CREB1 mRNA by over 80% (*Figure 3c*, *Figure 3—figure supplement 1*). By contrast, our screen did replicate previous studies that found CLK1 and RAB11FIP3 to influence viral infection (*Bruce et al., 2010*; *Karlas et al., 2010*; *Li et al., 2010*).

## Alternative splicing of host genes alters IAV replication

Having shown that many of the genes which are alternatively spliced during infection impact IAV replication, we next wanted to test whether the viral-induced alternative splicing of these genes impacted viral infection. In other words, in contrast to the experiments in *Figure 3* in which the complete gene was targeted, we sought to test the functional impact of specifically blocking expression of the viral-induced isoform. To do so, we employed a previously established approach in which antisense morpholino oligonucleotides (AMO's) are targeted to the exon-intron boundaries of alternative exons, promoting the skipping of the targeted exons (*Fiszbein et al., 2019*; *Mallory et al., 2011*; *Martinez et al., 2015*). By employing this strategy, splicing events where splice site usage is increased upon viral infection can be blocked, inhibiting expression solely of the isoform that is favored upon infection. Importantly, this strategy does not impact translation of the target mRNAs, as the AMOs only interact with the pre-mRNA. Of the 30 alternatively spliced genes shown to alter viral replication (*Figure 3b–d*), 11 exhibited increased exon inclusion upon IAV infection. Of these 11, we found that exon inclusion for five could be successfully modulated by AMO's: CLK1, RAB11-FIP3, PPIP5K2, IQCB1, and REPIN1 (*Figure 4a*, *Figure 4—figure supplement 1*). We also were able to modulate CREB1 splicing (*Figure 4—figure supplement 1*), which we included given the previous reports implicating this gene in IAV infection (*König et al., 2010*), as well as ERI2, which is a case of alternative 3' splice site that alters the C-terminus of the protein (*Figure 4a*, *Figure 4—figure supplement 1*).

To assess if AMO treatment influenced IAV infection or IAV titers, A549 cells transfected with AMO's were infected with IAV at a moi of 0.01 for 48 hr. Levels of viral replication were measured by RT-qPCR assaying vRNA levels (*Figure 4b*). In addition, infectious particles were measured via plaque assay (*Figure 4c*). CREB1, IQCB1, and REPIN1 AMO treatment resulted in little to no change at the viral titer or vRNA level compared to mock treatments (*Figure 4b,c*). However, remarkably, switching the splicing of ERI2, CLK1, RAB11FIP3, and PPIP5K2 by AMO resulted in significant changes of at least twofold at both the titer and vRNA level (*Figure 4b,c*) with no general toxicity to the cells (*Figure 4—figure supplement 1*). Notably, forced skipping of the CLK1 variable exon is predicted to have the same impact at siRNA treatment, as the exon-skipped product is predicted to be targeted for NMD (*Figure 4—figure supplement 1*). By contrast, both the exon-included and -skipped isoforms of RAB11FIP3, PPIP5K2, and ERI2 are predicted to encode active proteins (*Figure 4—figure supplement 1*). The results from the AMO experiment strongly suggest that the exon-included isoform of RAB11FIP3 and PPIP5K2 specifically promotes viral growth and/or that the exon-skipped product inhibits viral replication. Similarly for ERI2, viral infection normally favors the shorter isoform, while inhibition of this isoform in favor of the longer version markedly inhibits viral infection and replication. Although the impact of splicing modulation of CLK1, RAP11FIP3 and PPIP5K2 are relatively modest, given that several splicing changes occur at once during IAV infection, we also tested whether modifying multiple splicing changes simultaneously would have additive effects. Indeed, a combination AMO treatment targeting CLK1, RAB11FIP3, and PPIP5K2 splicing together, resulted in a larger decrease in viral titer compared to any of these single AMO treatments (*Figure 4d*; ~fourfold instead of ~twofold). Together, these data show that direct alteration of host splicing events upon viral infection impacts viral replication, at least for a subset of genes. Even for those genes in which we do not observe a direct functional impact of viral-induced splicing (e.g. IQCB1), it is possible that splicing regulation contributes to gene function and viral replication in a manner that is beyond the detection of our current assays.

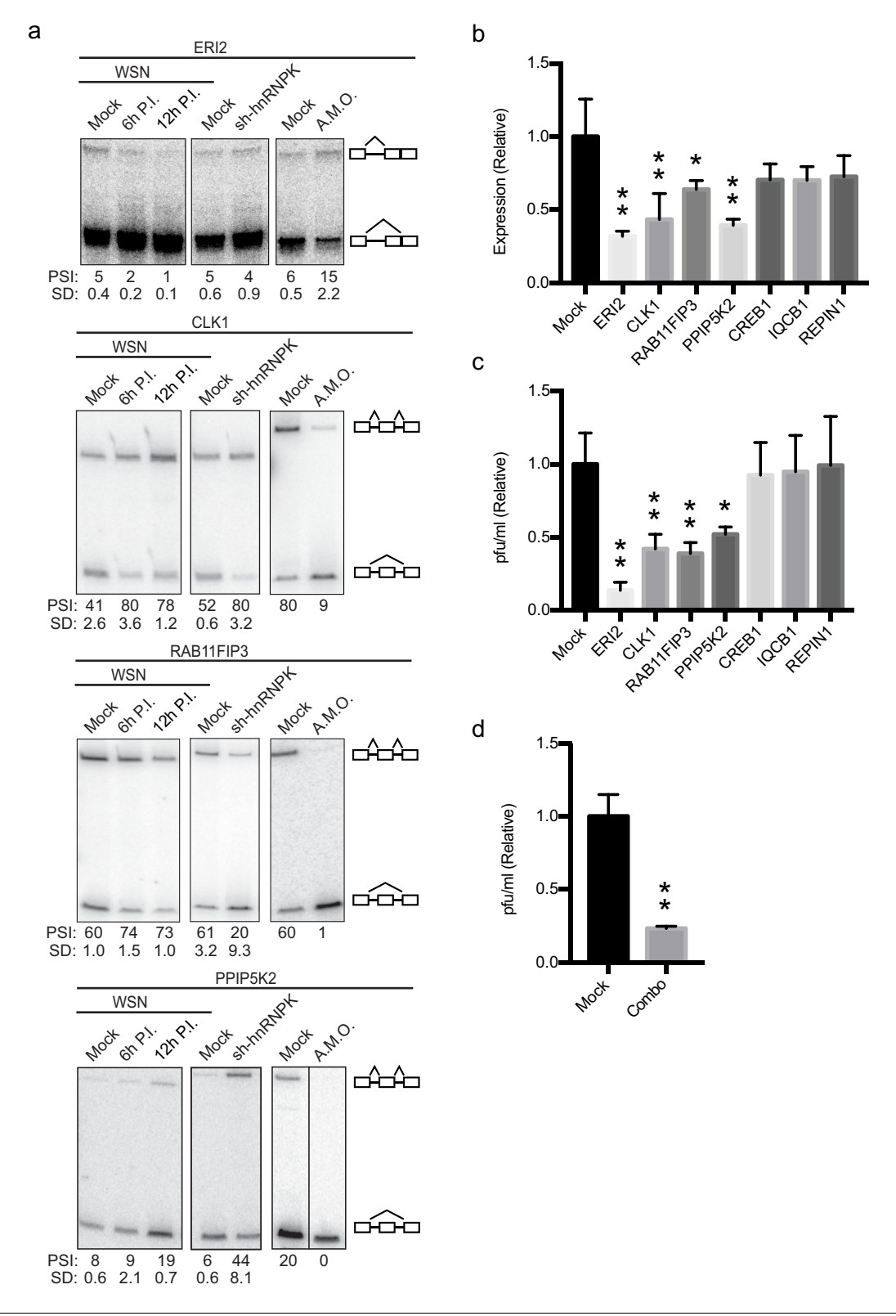

**Figure 4.** Altering splicing of host genes results in IAV replication changes. (a) A549 cells were infected (WSN), treated with shRNA against hnRNP K, or treated with splice-blocking antisense morpholino oligonucleotides (AMOs). Splicing was analyzed using RT-PCR analyses (b) RT-qPCR analyses (values normalized to mock AMO treatment) of PA segment vRNA 48 HPI with IAV (moi = 0.01) in the context of AMO treatment (targeted genes indicated on x-axis). Values are means ± s.d. from three independent experiments. (c) Measurement of plaque-forming units (values normalized to mock AMO

*Figure 4 continued on next page*

*Figure 4 continued*

treatment) in media 48 HPI with IAV WSN (moi = 0.01) in the context of AMO treatment (targeted genes are indicated on x-axis). Values are means ± s.d. from three independent experiments. (d) Measurement of plaque-forming units (values normalized to mock AMO treatment) in media 48 HPI with IAV WSN (moi = 0.01) in the context of AMO treatment. In combo-treated cells CLK1, RAB11FIP3, and PPIP5K2 were simultaneously targeted with AMO's. Values are means ± s.d. from three independent experiments. Statistical significance in (b) and (c) was determined via one-way ANOVA with multiple comparisons to mock corrected via Dunnett's test and in (d) via two-tailed student's t-test. p<0.05 = * and p<0.005 = ** when comparing mock AMO treatments vs AMO treated samples.

The online version of this article includes the following figure supplement(s) for figure 4:

**Figure supplement 1.** AMO-induced alternative splicing of selected genes.

## hnRNP K regulates a subset of influenza A sensitive splicing events

We next set out to investigate how IAV infection impacts alternative splicing. We focused on the host protein hnRNP K because it was previously shown to be a pro-viral factor that regulates splicing of IAV M segment RNA (*Mor et al., 2016*; *Thompson et al., 2018*; *Tsai et al., 2013*). We and others have also shown that hnRNP K regulates splicing of host transcripts (*Thompson et al., 2018*; *Venables et al., 2008*). Therefore, we first tested if hnRNP K is important for IAV replication independent of its regulation of M segment splicing. In previous work, we generated an IAV mutant strain in which hnRNP K cannot bind to the M segment to regulate splicing (ΔK-mut; *Thompson et al., 2018*). To determine if hnRNP K regulates IAV replication in the context of this virus, we depleted hnRNP K in A549 cells using siRNAs and assayed infectious virus particles 48 hr post-infection. Importantly, we show that while the ΔK-mut IAV M2 splicing is no longer sensitive to hnRNP K levels, viral propagation is still attenuated upon hnRNP K knockdown (*Figure 5a*, *Figure 5—figure supplement 1*). These data indicate that hnRNP K is required for IAV replication independent of its role in M splicing.

To determine if any of the IAV-sensitive alternative splicing events are dependent on hnRNP K, we depleted hnRNP K in A549 cells by stably expressing a doxycycline-inducible shRNA (*Figure 5b*). Splicing changes that occur in response to hnRNP K knockdown were quantified using MAJIQ, filtering for significant changes of 10% or more in percent splicing (ΔPSI). In agreement with previous studies, we observe large changes in host splicing upon hnRNP K depletion (*Supplementary file 2*; *Thompson et al., 2018*; *Venables et al., 2008*). Importantly, 21% of the influenza-induced splicing changes are also hnRNP K-responsive (*Figure 5c*). To determine if hnRNP K positively or negatively regulates splicing of the IAV-dependent splicing events, we plotted the impact of hnRNP K and IAV infection on PSI values of the 179 common splicing events (*Figure 5d*). We find 65% of splicing changes are regulated similarly by hnRNP K depletion and IAV infection, whereas for the remainder, hnRNP K depletion has the opposite effect of infection. We validated many of these splicing changes in response to both IAV infection and hnRNP K knockdown by RT-PCR, including both those that exhibit a similar and an opposite change under these two conditions (*Figure 4a*, *Figure 2—figure supplement 1*). The observed high validation rates are consistent with previous studies with MAJIQ (*Vaquero-Garcia et al., 2016*). In addition to hnRNP K-induced regulation of splicing, we also find that hnRNP K alters the expression of ~2000 genes, some of which are also altered upon viral infection (*Figure 5—figure supplement 2*, *Supplementary file 1*). These hnRNP K regulated genes are generally independent of hnRNP K regulated splicing events (8% of splicing events are differentially expressed, *Figure 5—figure supplement 2*). Some of these changes in gene expression may contribute to regulation of IAV infection. However, as there is no enrichment for viral-related functions among the hnRNP K-induced differentially expressed genes (*Figure 5—figure supplement 2*), and our focus here is on splicing. We have not further pursued the impact of this differential expression.

## hnRNP K nuclear distribution is altered during infection

Based on the two populations of hnRNP K and IAV-responsive splicing events, we propose at least two models for how hnRNP K-dependent splicing is regulated during infection. The abundance of splicing events for which hnRNP K depletion phenocopies infection at the splicing level suggests that hnRNP K may be functionally depleted or inhibited during infection. By contrast, for events with opposite ΔPSI values between hnRNP K depletion and infection, hnRNP K activity may be enhanced

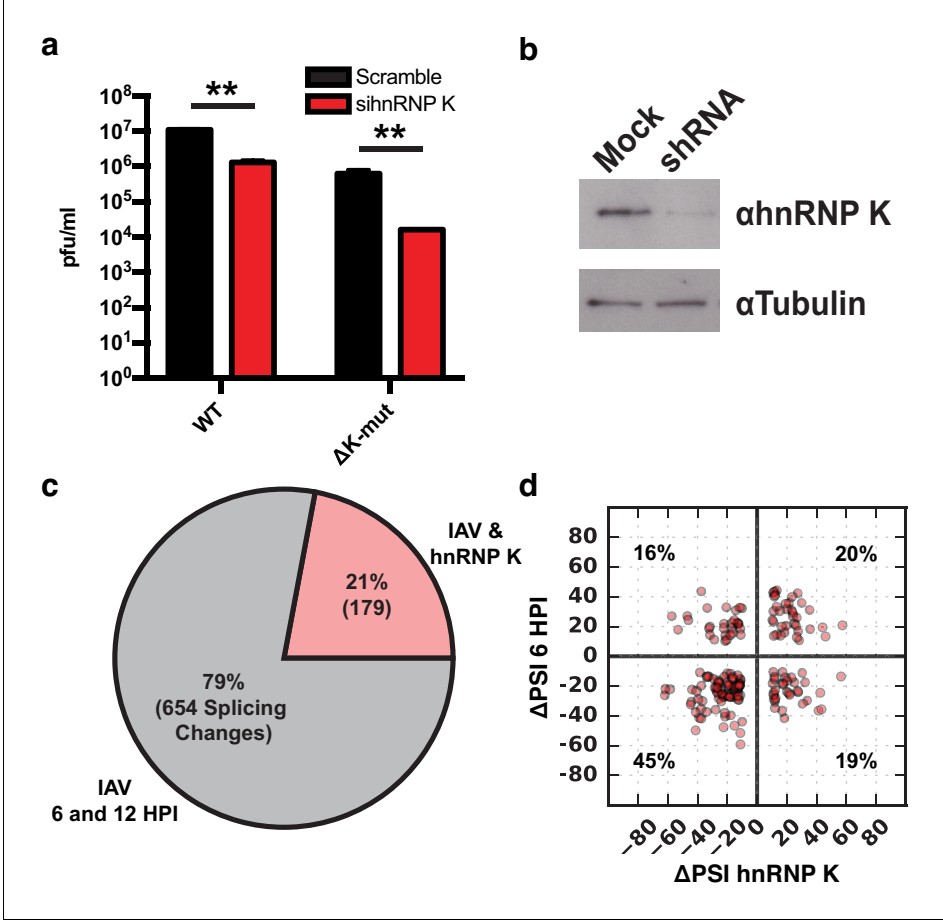

**Figure 5.** hnRNP K regulates a subset of IAV-induced splicing events during infection. (**a**) A549 cells were infected with WT and ΔK-mut IAV in the context of scrambled or hnRNP K targeting siRNA. Plaque-forming units in media were assayed at 48 HPI. Values are means ± s.d. from three independent experiments. p<0.005 = **. Statistical significance was determined via two-tailed student's t-test where p<0.05 = * and p<0.005 = ** when comparing scramble vs siRNA-treated samples. (**b**) Western blot of shRNA-treated cells. Tubulin is shown as a loading control. (**c**) Proportion of IAV-sensitive splicing events (observed at 6 and 12 HPI) that are also sensitive to hnRNP K knockdown. (**d**) ΔPSI value comparison of 179 splicing events at 6 HPI or after hnRNP K knockdown. Percentage values denotes proportion of 179 splicing events residing in each quadrant.

The online version of this article includes the following figure supplement(s) for figure 5:

**Figure supplement 1.** IAV M segment splicing in hnRNP K-depleted cells.
**Figure supplement 2.** General analysis of gene expression in hnRNP K-depleted A549 cells.

---

by a co-factor that is increased or activated upon infection. Both these models imply a change to hnRNP K activity upon infection. However, western blot analysis revealed little to no differences in hnRNP K expression or nucleocytoplasmic localization (*Figure 6a*). Analyses of hnRNP K phosphorylation via phos-tag SDS-PAGE also showed no reproducible changes overall, although we do note a minor increase in phosphorylation, particularly in the nuclear fraction (*Figure 6a*). These results are in agreement with previous mass spectrometry-based studies showing that total hnRNP K level and phosphorylation do not significantly change during IAV infection (*Coombs et al., 2010*; *Dapat et al., 2014*).

Although hnRNP K levels did not appear to change between the cytoplasm and nucleus, it is possible more subtle changes in hnRNP K subcellular localization occur during infection. In past work, we showed that hnRNP K activates IAV M RNA splicing within nuclear speckles, as opposed to the nucleoplasm where most splicing of host transcripts is thought to occur (*Mor et al., 2016*; *Thompson et al., 2018*). Therefore, we asked if the speckle-associated pool of hnRNP K is altered

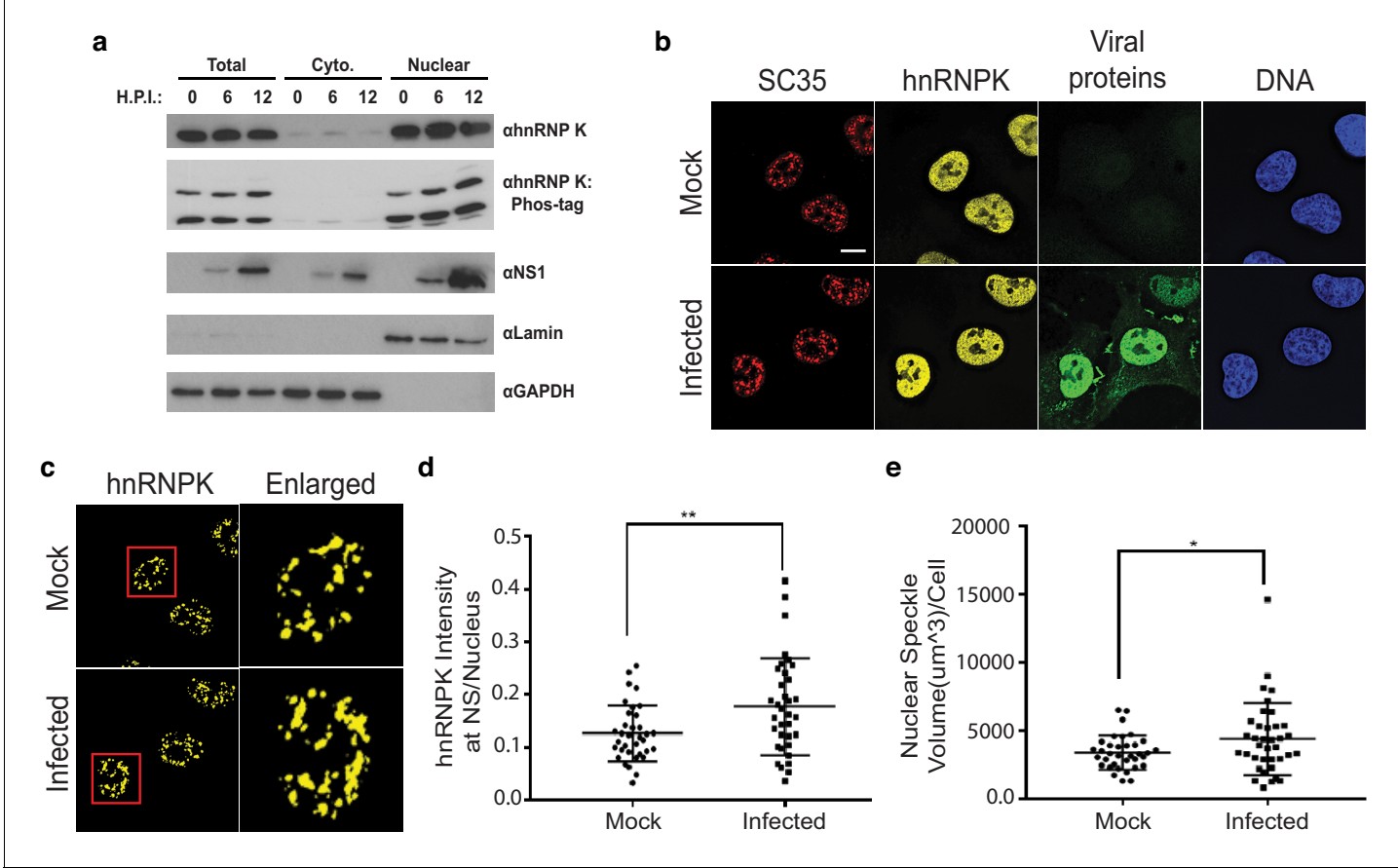

**Figure 6.** hnRNP K levels increase at nuclear speckles during influenza A virus infection. (**a**) Western blot analyses of total, cytoplasmic, or nuclear A549 cell extracts 0, 6, and 12 hr post infection (HPI). hnRNP K was analysed via traditional SDS-PAGE or Phos-tag to separate phosphorylated isoforms of hnRNP K. (**b**) Approximately 100% of A549 cells were infected with influenza virus (WSN) for 5 hr and then subjected to immunofluorescence microscopy to detect SC35, hnRNP K, and viral proteins. Nuclear speckles were marked with anti-SC35 antibody. Scale bar = 10 µm. (**c**) Voxel outside surface generated for nuclear speckles was set to 0 for hnRNP K channel to visualize hnRNP K at nuclear speckles. The marked region is enlarged. (**d**) Quantification of hnRNP K at nuclear speckles (NS) was performed using the Imaris software. hnRNP K intensity sum at nuclear speckles was normalized to the intensity sum in the nucleus. Values are means ± s.d. measured in 35 mock and 35 WSN infected cells from biological triplicates. ** unpaired, two-tailed t-test p<0.01. (**e**) Quantification of nuclear speckle volume was performed using the Imaris software. Surface was generated using fluoresce signal from immunostained SC35 proteins to measure nuclear speckle volume. Values are means ± s.d. measured in ≥35 mock and WSN infected cells from biological triplicates. * unpaired, two-tailed t-test p≤0.05.

The online version of this article includes the following figure supplement(s) for figure 6:

**Figure supplement 1.** RT-PCR analyses of CLK1 or CREB1 alternative splicing in Mock or IAV infected cells pretreated with siCtrl or sihnRNP K siRNA.

during IAV infection. To test this, hnRNP K colocalization with nuclear speckle-associated protein, SC35, was measured using fluorescence microscopy in mock and IAV-infected A549 cells (*Figure 6b–c*). Strikingly, we observed that hnRNP K levels at nuclear speckles increase modestly but significantly during IAV infection (*Figure 6d*). This increase in localization of hnRNP K to speckles was mirrored by an overall increase in speckle size upon infection (*Figure 6e*). Therefore, we propose that at least one of the mechanisms by which IAV results in alterations in hnRNP K-regulated splicing events is through enhanced recruitment of hnRNP K to nuclear speckles. Although this re-localization may not be sufficient to have a gross impact on the overall levels of hnRNP K in the nucleoplasm, it is possible that this re-localization, coupled with competition for hnRNP K between IAV and host RNAs, effectively depletes a particular pool of hnRNP K from co-transcriptional splicing processes in the nucleoplasm, or redirects hnRNP K activity to splicing events that occur at nuclear speckles (see Discussion). Regardless of the mechanism, we do find that upon depletion of hnRNP K from cells, there is no further change in the splicing of hnRNP K- and IAV-dependent events upon

IAV infection (*Figure 6—figure supplement 1*). While we cannot rule out that the lack of IAV-induced splicing is an indirect result of reduced infection efficiency, these results are consistent with our model that reduced hnRNP K activity is the primary driver of a subset of IAV-induced host splicing.

## Discussion

Previous studies have established that the host transcriptome is profoundly altered upon IAV infection. While much of this work has shown that regulation of host transcription is a major component of these changes, only recently have post-transcriptional events, such as alternative splicing, begun to be analyzed (*Fabozzi et al., 2018*). Here, we identify a program of host splicing that is regulated by IAV infection and show that many of these splicing changes occur within genes that promote IAV replication. Furthermore, directly manipulating a subset of these IAV-induced splicing changes, using splice-blocking AMO's, also attenuates infection. Together these data implicate infection-induced changes in host splicing as a previously unrecognized mechanism by which viruses promote infection.

Our RNA-seq analyses of A/WSN/33-infected A549 cells revealed functionally distinct categories of genes that are regulated by transcriptional and splicing responses. Genes that exhibit differential expression are enriched for genes involved in immune response and viral defense, consistent with previous studies (*Fabozzi et al., 2018*; *Ioannidis et al., 2012*). By contrast, GO analyses of genes that harbor splicing changes demonstrates enrichment for RNA processing categories. Moreover, we observe very little overlap between the genes that are differentially spliced versus those that show differential expression. These data suggest that the host splicing response to virus is functionally distinct from the transcriptional response. We are unable to directly compare our data to other RNA-Seq studies due to differential experimental design and depth of sequence; however, our findings are generally consistent with the prediction that IAV infection induces changes in host splicing in addition to, and separate from, changes in transcription. Patterns similar to this have been observed in other systems. For example, T-cell stimulation results in robust transcriptional and splicing changes; however, very little overlap is observed between the categories (*Martinez et al., 2012*).

Our analyses of host splicing changes adds to a pattern of IAV inducing a variety transcriptomic phenotypes. Recent studies observing altered genome architecture, transcript 3′ end defects, and PA-X targeted transcript degradation during infection all have links to splicing (*Gaucherand et al., 2019*; *Heinz et al., 2018*; *Zhao et al., 2018*). One observation was that of transgenic, defective, and unannotated splicing in relation to transcription termination (*Heinz et al., 2018*; *Zhao et al., 2018*). While our splicing analyses did not account for these specific splicing types, we would predict for them to be present in our experiments to some extent. Importantly, the alternatively spliced transcripts we describe here are not products of defective or transgenic splicing, and are likely not induced by the same transcription defects described in other studies. Additionally, the IAV endonuclease PA-X was shown to preferentially target spliced transcripts (*Gaucherand et al., 2019*). However, the alternatively spliced genes we define here do not overlap with PA-X-targeted genes (*Gaucherand et al., 2019*). Therefore, while splicing appears to contribute to many facets of IAV-induced transcriptome changes, we propose that a portion of the functional splicing events described here are independent of previously described mechanisms.

Although the set of genes that we identify to undergo alternative splicing upon IAV infection are not enriched for known regulators of viral replication, we demonstrate that half of the splicing-regulated genes we tested (30/61) do impact IAV infection or propagation. This includes genes involved in cell signaling, intracellular trafficking and gene expression. In addition to demonstrating that many of the host genes that undergo IAV-induced splicing changes are involved in controlling infection, in several cases we directly show that changing the splicing of these genes alters IAV replication (*Figure 4*). For example, IAV infection induces inclusion of alternative CLK1 exon 4, which results in expression of the canonical full-length isoform of the protein (*Figure 4a*). By contrast, skipping of exon 4 is observed in uninfected cells and introduces an early stop codon that truncates the kinase domain of protein (*Duncan et al., 1997*; *Uzor et al., 2018*). Therefore, in the case of CLK1, both the AMO that blocks exon 4 inclusion (*Figure 4*), and the siRNA (*Figure 3*), are expected to decrease the expression of active protein. Given that previous studies have established CLK1 to be required

for efficient IAV replication (*Artarini et al., 2019*; *Karlas et al., 2010*; *Zu et al., 2015*), our data is consistent with a model in which IAV infection promotes exon inclusion to increase production of full-length functional CLK1 kinase that, in turn, promotes viral growth.

For RAB11FIP3, IAV induces the inclusion of a non-canonical in-frame alternative exon (*Figure 4a*), resulting in expression of an alternate isoform of this protein. The function of the additional protein sequence remains unclear, as it lies upstream of the defined RAB11-binding domain (*Shiba et al., 2006*) and does not contain known phosphorylation sites (*Collins et al., 2012*). Regardless, this non-canonical isoform appears to promote IAV infection, as blocking the variable exon upregulates the canonical isoform and limits IAV replication in multi-cycle infections (*Figure 4b*). Interestingly, a previously established model suggests that RAB11FIP3 may compete with IAV vRNPs for RAB11 on endosomes in the particle assembly pathway (*Vale-Costa et al., 2016*). Therefore, it is possible that the canonical RAB11FIP3 isoforms may interfere with particle assembly, such that it is beneficial to the virus for this canonical isoform to be decreased through splicing.

PPIP5K2 exon inclusion is also increased upon viral infection, producing increased levels of the canonical full-length isoform of the RNA (*Figure 4a*). By contrast, forced skipping of this exon results in decreases in viral titers and vRNA (*Figure 4a–c*). Interestingly, PPIP5K2 has been previously thought to limit to influenza A replication (*Pulloor et al., 2014*), as it was shown that catalytic activity of PPIP5K2 is required for efficient IFNβ production, and in turn, that knockdown of PPIP5K2 resulted in viral load increases. The exon that is skipped in uninfected cells removes a portion of peptide sequence without altering the downstream reading frame. It is possible that the resultant smaller protein has increased catalytic activity, perhaps through removal of an inhibitory interaction, such that altering the splicing of PPIP5K2 may reduce PPIP5K2 function.

ERI2 is a particularly interesting example of IAV-induced alternative splicing. The ERI2 gene encodes a poorly characterized exonuclease that is a member of a family of non-specific 3′ to 5′ exonucleases that target double-stranded RNA or DNA (*Dominski et al., 2003*). Use of an alternative 3′ splice site in the terminal exon results in a swap between two unique C-termini. The distal 3′ splice site, which is favored upon IAV infection, results in a 35 amino acid terminus, while use of the proximal 3′ splice site adds an additional 392 amino acids. Both isoforms contain the exonuclease domain, however the longer form also contains a putative nucleic-acid-binding zinc finger. While the functional role of this protein is unknown, we do note that it was identified in a broad siRNA screen as required for efficient HIV infection (*Brass et al., 2008*). Thus, the activity of this protein, its relevance for viral infection, and the functional difference between isoforms, will be an interesting area for future study.

Importantly, we show a previously established pro-viral host factor, hnRNP K, regulates a subset of the splicing events that change in response to IAV infection, including CLK1, RAB11FIP3, and PPIP5K2. Notably, hnRNP K undergoes increased sequestration into nuclear speckles upon viral infection. While the overall percentage of hnRNP K in the speckles remains low even after infection, given the plethora of targets of hnRNP K in the nucleus, including the IAV M transcript, it is reasonable that the redistribution of hnRNP K upon infection may alter the pool of free hnRNP K available to bind to and regulate host genes that are typically spliced outside of speckles. Further consistent with a functional role of hnRNP K in regulating important pro-viral host splicing events, while hnRNP K was previously established to be important for IAV M segment splicing regulation (*Mor et al., 2016*; *Thompson et al., 2018*; *Tsai et al., 2013*), IAV replication is sensitive to hnRNP K expression even in the context of a mutant IAV in which hnRNP K is not required for M segment splicing regulation (*Figure 5a*, *Figure 5—figure supplement 1*). It is possible, however, that hnRNP K is also important for additional viral processes such as regulating transcription or translation of viral transcripts, as hnRNP K has been implicated to regulate many facets of RNA processing other than splicing (*Bomsztyk et al., 2004*). Nonetheless, given that hnRNP K levels regulate 21% of detected IAV-induced splicing changes (*Figure 5c*), many of which encode for proteins that influence viral replication (*Figure 3c*), we contend that hnRNP K regulation of host transcript splicing is an important process during IAV infection.

While our data clearly shows changes in host splicing upon IAV infection (*Figure 2*), many of which are regulated by host splicing factor hnRNP K (*Figure 5c*), it is not clear if these splicing changes are induced through host or viral mechanisms. A virus-centric model would suggest that viral manipulation of host splicing-related pathways causes the splicing changes observed during infection. By contrast, a host-centric model would predict the splicing changes observed during

infection are part of a host response to pathogen. Likely, the changes we observe here are a mixture of both models. However, RT-PCR analysis of a panel of IAV-sensitive splicing events in the context of IFNβ treatment identified no significant splicing changes, thus ruling out a model in which splicing changes are primarily triggered indirectly by interferon secretion, as a secondary consequence of the innate immune response.

In sum, we demonstrate here a feedback loop between viral-induced regulation of splicing of host genes and subsequent viral infection. This is yet another example of the intricate competition between host and virus, involving multiple layers of regulation and counter-regulation[53,54]. We also demonstrate one of the mechanisms driving this IAV-induced regulation of splicing. Future studies will be required to further uncover additional underlying mechanisms by which IAV control host splicing, and to determine if manipulation of these mechanisms has potential for antiviral therapeutic intervention.

## Materials and methods

### Cell culture and viruses

Human lung adenocarcinoma epithelial cells (A549) were cultured in RPMI 1640 (Corning: 10–040-CV), 10% heat-inactivated FBS (Gibco: 16000–044), and 100 units ml−1 Pen/Strep antibiotics. MDCK cells were cultured in high-glucose DMEM (Corning: 10–013-CV), 10% heat-inactivated FBS (Gibco) and 100 units ml−1 Pen/Strep antibiotics. All cells were maintained at 37°C with 5% $CO_2$. Cells were tested negative for mycoplasma. Cell lines were obtained and authenticated by ATCC (A549: CCL-185, MDCK: PTA-6500). A/Puerto Rico/8/1934 (PR8) and A/WSN/33 (WSN) strains were used as specified for individual experiments. ΔK-mut virus was generated as previously described in another study (*Thompson et al., 2018*). Viral titers were determined via plaque assay in MDCK cells as the average to technical replicates.

### Infections

For infections, A549 cells were grown to 80% confluency, washed with PBS, and inoculated with viral titer diluted in PBS•BA (DPBS with Ca and Mg (Corning: 21–031-CV), 0.2% BSA (Lampire: 7500810), and 100 units ml−1 Pen/Strep antibiotics) for 1 hr at room temperature (22°C). Cells were then washed with PBS and incubated in Infection media (1X MEM, 0.2% BSA, 10 mM HEPES buffer, 0.12% NaHCO$_3$, 100 units ml−1 Pen/Strep antibiotics) at 37°C with 5% $CO_2$ until desired time-point. At time-points, media was collected for plaque assay analyses. For protein and RNA analyses, wells were rinsed with PBS and either pelleted and lysed with RIPA buffer for western blot analysis or RNA was extracted using RNA-Bee (amsbio: CS-501B).

### Plaque assays

MDCK cells were grown to 100% confluency in 12-well plates. Viral titers to be assayed were serially diluted in PBS•BA (DPBS with Ca and Mg (Corning: 21–031-CV), 0.2% BSA (Lampire: 7500810), and 100 units ml−1 Pen/Strep antibiotics). MDCK cells were then washed in PBS and viral dilutions were added to each well and left to inoculate for 1 hr at room temperature (22°C). Viral dilutions were then removed and 1 ml of plaque overlay media was added to each well (1X MEM, 0.2% BSA, 10 mM HEPES buffer, 0.22% NaHCO$_3$, 0.01% DEAE Dextran), 100 units ml−1 Pen/Strep antibiotics, and 0.6% Agar (Oxoid). Plates were stored upside-down at 37°C with 5% $CO_2$ for 48 hr at which point cells were fixed with 3.6% formaldehyde. Gel plugs were then removed, cells were stained with Crystal violet, and plaques were counted.

### Depletion of hnRNP K from A549 cells

For knockdown of hnRNP K to test viral replication (*Figure 5a*), A549 cells were treated with 50 nM siRNA (Dharmacon, SMARTpool, M-011692–00) as previously described (*Thompson et al., 2018*). For RNA-Seq and validation experiments, hnRNP K was stably depleted from A549 cells using a lentiviral expressed doxycycline-inducible shRNA with sequence: agcgGGACCTATTATTACTACACAATAGTGAAGCCACAGATGTATTGTGTAGTAATAATAGGTCC (target underlined) This shRNA vector and its use are described in detail previously (*Martinez et al., 2015*).

### RNA-seq of infected or hnRNP K knockdown cells

For infections A549 cells were grown to 80% confluency in 15-cm plates and infected with WSN IAV at moi = 2. At the 0, 6, and 12 hr post infection, total RNA was extracted from cells. For hnRNP K knockdowns, either stable shRNA expressing or WT A549 cells were grown to 90% confluency in 15-cm plates over a span of 48 hr in the presence of doxycycline. At 48 hr total RNA was extracted. RNA was then DNase treated and further purified using RNeasy mini kit (Qiagen). RNA quality assessed for quality using a Nanodrop and Agilent 2100 Bioanalyzer. Samples were required to have a 260/280 ratio of >1.8 as well as a RIN value of >6 to be used for library preparation. Poly(A) selected cDNA libraries were generated on site at the University of Iowa sequencing core using standard methods. cDNA libraries were then sequenced for paired-end, 150-nt, reads on an Illumina HiSeq 4000. RNA-Seq data is accessible in GEO under study GSE142499.

### RNA-seq mapping, differential expression analysis

Raw reads were aligned to the human or viral WSN genome using STAR (*Dobin et al., 2013*). Differential gene expression of host transcripts was performed using the R-package, DESeq2 (*Love et al., 2014*). Significantly changing genes were defined as those with a log2 fold change greater than |1| and a corresponding p-value<0.05.

### Splicing analysis

Spicing analyses was performed using MAJIQ (ver. 1.1.3a) as described previously (*Vaquero-Garcia et al., 2016*). Initial splicing analyses of 0 HPI, 6 HPI, 12 HPI, mock dox-treatment, and sh-hnRNP K samples were performed in parallel using MAJIQ builder with default settings. ΔPSI values were quantified with the MAJIQ package, VOILA, using the following settings. Six HPI samples were quantified against 0 HPI set at a threshold of 10. Twelve HPI samples were quantified against 0 HPI set at a threshold of 20. sh-hnRNP K samples were quantified against mock dox-treated samples at a threshold of 10. Significant splicing events were those with probability values > 0.95 and a ΔPSI value greater than 10 (for 6 HPI and sh-hnRNP K) or 20 (for 12 HPI).

### siRNA screen

The siRNA screen was done as previously described (*Panda and Cherry, 2015*). In brief, siRNA libraries targeting each gene were ordered from Ambion and aliquoted into 384-well plates. siRNAs were then transfected into 2000 A549 cells in each well using 0.5 µl HiPerFect (Qiagen) in a total volume of 50 µl and a final concentration of 20 nM siRNA. siRNA treatment proceeded for 72 hr. siRNA transfection efficiency was monitored using the siDeath control RNA. Cells were then infected with IAV PR8 at an moi of 0.1 or 0.5 for 24 hr. After infections cells were fixed with 4% formaldehyde, washed twice with phosphate-buffered saline (PBS), blocked for 1 hr in blocking solution consisting of PBS containing 2% bovine serum albumin (BSA), 0.1% Triton X-100, and 0.02% sodium azide, and stained with mouse anti-IAV NP primary antibody at a 1:2000 dilution in blocking solution overnight at 4°C. The primary antibody was then replaced with blocking solution containing 5 µg/ml bisBenzimide H 33342 trihydrochloride (Sigma-Aldrich, B2261) and goat anti-mouse Alexa Fluor 488 secondary antibody (Invitrogen, A11029) at a 1:1000 dilution and incubated for 1 hr at room temperature. The cells were then washed three times with PBS containing 0.1% Triton X-100 before imaging. Imaging of cells was performed using an ImageXpress autoscope and MetaXpress software. Four images were collected per well and a median % IAV cell score was calculated. Changes in % IAV positive cells were calculated relative to wells treated with siControl siRNA. Final values were determined as an average % change in IAV positive cells across duplicate experiments.

### Antisense morpholino oligonucleotide treatment and infections

For each AMO treatment, 500,000 A549 cells were electroporated with a total of 5 nmoles AMO using a Bio-Rad Gene Pulser Xcell (settings used were default A549 cell program). AMO sequences are detailed in *Supplementary file 3*. Electroporations were performed in RPMI media without FBS or antibiotics. 200,000 treated cells were seeded into 6-well plates and allowed to grow for 48 hr. Splice-blocking activity each AMO at 48 hr was validated using radio-labeled RT-PCR (*Figure 4a*, *Figure 4—figure supplement 1*). Cells were then infected with WSN IAV at the specified moi and

time-course. At the specified time points, media was collected for downstream plaque assays and total RNA was collected for further experiments.

## Microscopy and quantification of subnuclear localization of hnRNP K

For subnuclear localization of hnRNP K, immunofluorescence was performed as previously described (25) using SC35 (ab11826, Abcam), hnRNP K (GTX101786, Genetex) and anti-influenza A virion poly-clonal antibodies (B65141G, Meridian Life Science). For quantification of hnRNP K localization, images were captured as previously described (25). AutoQuant software was used to deconvolve z stack images. Deconvolved images were used for surface generation and quantification using Imaris (Bitplane) software. Surface was generated for nuclear speckles using the SC35 channel. Display minimum and maximum values were adjusted to distinguish SC35 at nuclear speckles from the nucleoplasm pool. Display minimum and maximum values were kept constant for all images. DAPI channel was used for generating the nuclear surface. Threshold was adjusted manually during surface generation. Number of voxels above 10 was used as filter. Intensity sum of hnRNPK from nuclear speckle surface and nucleus surface was quantified using the Imaris software. Volume of nuclear speckle was quantified using Imaris software. Statistical analysis was done using an unpaired, two-tailed, t-test with Welch's correction was performed.

## Interferon treatment

A549 cells were grown 80% confluency in 10-cm plates. At time of interferon treatment, fresh, pre-warmed media containing 10 ng/ml beta-interferon (STEMCELL Technologies, Cat #78113) was added to cells. Total cellular RNA was harvested 12 hr post treatment using RNA-Bee (amsbio: CS-501B). IFIT1 expression was assessed via qPCR (*Figure 2—figure supplement 1*) to validate innate immune stimulation.

qPCR analyses cDNA was produced in a 24 µl reaction using 500 ng total cellular RNA, 1 ng of gene-specific reverse primer, and MMLV reverse transcriptase enzyme. 3 µl of cDNA was added to 10-µl qPCR reactions containing 0.5 µM forward and reverse primers as well as 1X PowerUP SYBR green master mix (ThermoFisher) in 384-well plates. Expression levels of targeted genes were calculated relative to GAPDH. qPCR primers are detailed in *Supplementary file 3*. All qPCR data is from triplicate biologic replicates, each analyzed in technical duplicate.

## RT-PCR analyses

RT-PCR was carried out as described in detail previously (*Martinez et al., 2015*; *Thompson et al., 2018*). RT-PCR primers are detailed in *Supplementary file 3*. All RT-PCR data is from analysis of triplicate biologic replicates.

## Acknowledgements

We thank members of the Lynch, Cherry and Fontoura laboratories for helpful comments throughout this study. The NS1 antibody was a gift from Dr. Adolfo Garcia-Sastre. This work was funded by NIH grants R01 AI125524 to KWL and BMAF and R35 GM118048 to KWL. SC was supported by grants from the NIH (R01AI150246, R01AI122749, R01AI140539) and is a recipient of the Burroughs Wellcome Investigators in the Pathogenesis of Infectious Disease Award.

## Additional information

### Funding

| Funder | Grant reference number | Author |
|---|---|---|
| National Institutes of Health | R01 AI125524 | Matthew G Thompson<br>Prasanna Bhat<br>Beatriz MA Fontoura<br>Kristen W Lynch |
| National Institutes of Health | R35 GM118048 | Matthew G Thompson<br>Michael J Mallory<br>Max B Ferretti<br>Kristen W Lynch |

| National Institutes of Health | R01 AI150246 | Mark Dittmar Sara Cherry |
| National Institutes of Health | R01 AI122749 | Mark Dittmar Sara Cherry |
| National Institutes of Health | R01 AI140539 | Mark Dittmar Sara Cherry |

The funders had no role in study design, data collection and interpretation, or the decision to submit the work for publication.

## Author contributions

Matthew G Thompson, Conceptualization, Data curation, Formal analysis, Investigation, Methodology, Writing - original draft, Writing - review and editing; Mark Dittmar, Prasanna Bhat, Investigation, Writing - review and editing; Michael J Mallory, Investigation, Methodology, Writing - review and editing; Max B Ferretti, Formal analysis, Investigation, Visualization, Methodology, Writing - review and editing; Beatriz MA Fontoura, Sara Cherry, Conceptualization, Supervision, Writing - review and editing; Kristen W Lynch, Conceptualization, Formal analysis, Supervision, Funding acquisition, Writing - original draft, Writing - review and editing

## Author ORCIDs

Matthew G Thompson (iD) https://orcid.org/0000-0003-3813-9254
Kristen W Lynch (iD) https://orcid.org/0000-0002-0120-8079

## Decision letter and Author response

Decision letter https://doi.org/10.7554/eLife.55500.sa1
Author response https://doi.org/10.7554/eLife.55500.sa2

# Additional files

## Supplementary files

• Supplementary file 1. Differential Expression of host genes in A549 cells infected with influenza A virus or depleted of hnRNP K. Fold change (log2) and p-values for genes are given at 6 and 12 hr post infection with IAV (page 1 and 2, DE_6HPI and DE_12HPI), or after depletion of hnRNP K by shRNA (page 3, DE_shHNRNPK).

• Supplementary file 2. Splicing changes of host genes in A549 cells infected with influenza A virus A or depleted of hnRNP K. Change in splicing (delta percent spliced isoform or dPSI) for specific spliced junctions are given at 6 and 12 hr post infection with IAV (page 1 and 2, WSN_6HPI and WNS_12HPI), or after depletion of hnRNP K by shRNA (page 3, shHNRNPK).

• Supplementary file 3. Primers used in this study. Sequences are given for siRNAs for screen (page 1), primers for RT-PCR (page 2), splice-blocking AMOs (page 3), primers for qPCR (page 4), and primers for primer extension analysis of IAV (page 5).

• Transparent reporting form

## Data availability

Sequencing data have been deposited in GEO under accession code GSE142499.

The following dataset was generated:

| Author(s) | Year | Dataset title | Dataset URL | Database and Identifier |
|---|---|---|---|---|
| Thompson MG, Dittmar M, Mallory MJ, Bhat P, Ferretti MB, Fontoura BMA, Cherry S, Lynch KW | 2020 | Viral-Induced Alternative Splicing of Host Genes Promotes Influenza Replication | https://www.ncbi.nlm.nih.gov/geo/query/acc.cgi?acc=GSE142499 | NCBI Gene Expression Omnibus , GSE142499 |

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
