## [Decision Letter]

**Acceptance summary:**

We believe your study adds valuable insight into how viruses affect the biology of host cells by altering the pre-mRNA splicing apparatus to aid viral replication. This work also provides clues to the molecular mechanisms mediating these splicing changes. These results will be of general interest for many groups working in viral and RNA biology.

**Decision letter after peer review:**

Thank you for submitting your article "Viral-induced alternative splicing of host genes promotes influenza replication" for consideration by *eLife*. Your article has been reviewed by three peer reviewers, and the evaluation has been overseen by Douglas Black as Reviewing Editor and James Manley as the Senior Editor. The following individual involved in review of your submission has agreed to reveal their identity: Mariano A Garcia-Blanco (Reviewer #1).

The reviewers have discussed the reviews with one another and the Reviewing Editor has drafted this decision to help you prepare a revised submission.

Summary:

Thompson et al. examine host gene expression in A549 cells infected with Influenza A virus. Performing RNA sequencing and quantifying splicing changes they make the interesting observation that genes that were differentially regulated at the transcription level were mostly found in the antiviral response and DNA repair pathways while the genes that were differentially spliced were mostly found in splicing and mRNA transport pathways. They catalogued the types of splicing changes and focused on a set of splicing events for mechanistic studies. Using siRNAs to knockdown expression and antisense morpholino oligonucleotides (AMO) to drive splicing pattern changes, they determined the effects of particular genes and isoforms on influenza viral replication and production. Finally, the authors showed that hnRNPK is a regulatory factor for 21% of the altered splicing events during influenza virus infection. These events are possibly controlled by increased recruitment of hnRNP K to nuclear speckles during infection.

This study is one of the first to comprehensively quantify host splicing changes during viral infection. The authors clearly demonstrated effects on viral replication and production by modulating the splicing of some of these host transcripts, suggesting that these splicing events play an important role during viral infection. The authors also briefly mention in the Discussion that IFN treatment did not trigger similar splicing changes. Thus, the splicing changes may be an outcome of viral infection rather than a host response to pathogens.

All the reviewers found the paper to be interesting and important and the manuscript to be well written. However, they also agreed that the significance of the findings is weakened by the lack of method validation and essential controls. Moreover, several puzzling aspects of the results require more detailed explanation.

Essential revisions:

1) There is no data presented to confirm the knockdowns by either qRT-PCR or immunoblotting. Granted it is a significant amount of work to validate the 61 alternatively spliced candidate genes by siRNAs. However, without this data it impossible to assess possible correlations between the phenotype and the degree of knockdown. Given the complex phenotype assayed (viral infection) it is critical to consider off target effects of siRNAs. Although the experience of the collaborators suggests this was done properly, it should nonetheless be addressed explicitly in the study. The authors are no doubt aware that siRNA screens are plagued by high false positive rates. To manage these off-target effects, the authors need to validate the important hits using individual siRNAs. As better proof that the siRNAs are specific, the authors should consider using C911 siRNA controls (Buehler et al., PLoS ONE 2012) and determine whether these abrogate the transcript knockdowns and effects on viral replication. It would be worth testing these controls for the 5 genes (CLK1, RAB11FIP3, PPIP5K2, REPIN1 and IQCB1), for which the effects of splicing changes on viral replication and production were tested in Figure 4. Given that the effects seen from the siRNA screen (Figure 3B) were modest, and indeed the criterion for selecting hits at 20% inhibition was relaxed, it would be more convincing if all of the 61 factors or at least the most promising ones were additionally tested for decrease in formation of progeny viruses (as in Figure 3D).

2) The parallel discussion of the siRNA and AMO data was confusing and more in-depth description of the results is needed. Two of the genes (CLK1 and RAB11FIP3) that showed a significant difference in viral replication and production upon efficient AMO targeting are genes that showed the smallest reduction in viral replication upon siRNA knockdown. Without more information on the siRNAs and the efficiency of the knockdowns, it is difficult to draw conclusions about what the individual genes and splicing events might do. For example, do the CLK1 siRNAs target the exon that was skipped by AMO targeting and therefore both siRNA and AMO treatment gave the same viral phenotype? On the other hand, IQCB1 gave the greatest reduction in viral replication upon siRNA targeting but did not have any effects upon AMO targeting. What does this mean – perhaps that both isoforms are needed for viral replication? The functional consequences of the AS events could only be tested for 11 genes that showed increased exon inclusion on IAV infection. Of these 5 could be modulated (CLK1, RAB11FIP3, PPIP5K2, IQCB1, and REPIN1) – and of these only IQCB1 had an effect greater than 50% upon knockdown. While knockdown of all isoforms may not predict the effect of splice modulation, why not modulate the splicing of genes that showed the most dramatic effect upon knockdown (e.g., ERI2)? In cases where genes showed decreased exon inclusion, could ectopically expressing different isoforms affect viral replication?

3) The authors assert that hnRNP K accumulates to higher levels in nuclear speckles during IAV infection by performing immunofluorescence (IFA) (Figure 6). However, differences in hnRNPK localization +/- IAV was not evident in the IFA images provided. The quantification in Figure 6D suggests there are subtle differences, but these are not easily discerned upon visual inspection of Figure 6C. The authors should consider other means to quantify the immunofluorescence, perhaps by measuring the variance in the fluorescence across the nucleus. Another important control here is to perform a similar analysis on SC35 to assess the possibility that IAV infection increases the size of the nuclear speckles (and thus all proteins in them). Are these changes specific to hnRNPK? hnRNPK is highly abundant, so how to determine if a slight increase in its nuclear speckle concentration is functionally relevant? Do small changes in its overall concentration affect splicing? Some of these issues will be difficult to address, but the interpretations of the data should consider them.

4) The link between hnRNPK and splicing changes that occur during IAV infection is supported mostly by correlational data (Figure 5); direct evidence that hnRNPK activity is responsible for the IAV-induced splicing changes is lacking. A more direct link might be provided by analyzing a panel of genes that undergo similar isoform shifts during IAV infection and hnRNPK knockdown. Among such genes, does knock down of hnRNPK in IAV infected vs. uninfected cells lead to an additive effect on PSI changes or no further change (suggesting hnRNPK is the primary means by which IAV alters splicing)? Similarly, given that the ∆K-mut IAV appears to be more sensitive to hnRNP K depletion than wt IAV (Figure 4A), perhaps splicing modulation of certain genes would reveal a stronger phenotype in ∆K-mut infection.

5) The authors did not cite or discuss several papers that are pertinent to this study that have also profiled IAV-induced host gene expression changes at multiple levels (transcription, splicing, mRNA turnover, translation). Some examples: Bercovich-Kinori et al., 2016, performed RNA-Seq and ribosome profiling in IAV-infected cells, showing a prominent host shutoff phenotype. Bauer et al., 2018, used NETseq to study host transcription during IAV infection. Gaucherand et al., 2019, directly assessed the impact of IAV infection on the accumulation of spliced host transcripts, showing that the flu PA-X protein degrades spliced transcripts more efficiently than unspliced transcripts, and this is mediated through interactions between PA-X and proteins involved in splicing including hnRNPK. Some of these studies used PR8 rather than WSN, but WSN (because of its NS1-mediated targeting of Pol2) should have an even more prominent effect on host mRNA depletion. This previous work provides an essential background for the study presented here and should be discussed.

6) The authors contend that the changes in AS were not due to innate immune mechanisms because these changes were not seen with IFN treatment. The authors should consider that the innate immunity observed early in viral infection (in the case of flu 6-12 h pi) is best modeled by combined IFN treatment and dsRNA activation, not IFN alone. In future studies, it would also be interesting to test influenza virus mutants that are unable to antagonize the IFN response and see what types of splicing changes occur.

---

## [Author Response]

Essential revisions:1) There is no data presented to confirm the knockdowns by either qRT-PCR or immunoblotting. Granted it is a significant amount of work to validate the 61 alternatively spliced candidate genes by siRNAs. However, without this data it impossible to assess possible correlations between the phenotype and the degree of knockdown. Given the complex phenotype assayed (viral infection) it is critical to consider off target effects of siRNAs. Although the experience of the collaborators suggests this was done properly, it should nonetheless be addressed explicitly in the study. The authors are no doubt aware that siRNA screens are plagued by high false positive rates. To manage these off-target effects, the authors need to validate the important hits using individual siRNAs. As better proof that the siRNAs are specific, the authors should consider using C911 siRNA controls (Buehler et al., PLoS ONE 2012) and determine whether these abrogate the transcript knockdowns and effects on viral replication. It would be worth testing these controls for the 5 genes (CLK1, RAB11FIP3, PPIP5K2, REPIN1 and IQCB1), for which the effects of splicing changes on viral replication and production were tested in Figure 4. Given that the effects seen from the siRNA screen (Figure 3B) were modest, and indeed the criterion for selecting hits at 20% inhibition was relaxed, it would be more convincing if all of the 61 factors or at least the most promising ones were additionally tested for decrease in formation of progeny viruses (as in Figure 3D).

We appreciate the importance of these additional controls and have now added a new Figure 3—figure supplement 1 containing the requested data. The initial screen was actually done with two independent individual siRNAs. We apologize that this was not clear in the initial submission. We have now clarified this in the text and show in main Figure 3B the combined average of these individual siRNAs, while in Figure 3—figure supplement 1B we show the data for each of the individual siRNA. We have also added in Figure 3—figure supplement 1A qPCR data demonstrating knock-down for a representative subset of the 61 genes tested, including all of those which we follow in the rest of the study. In our hands, the C911 controls typically have a partial KD impact on target genes, which would complicate any interpretation. Therefore, we have instead included as controls for our KDs a scrambled siRNA (“NS”, Figure 3—figure supplement 1A) which does not impact viral infection (Figure 3), and the siRNA against hnRNP K, which does impact viral infection (Figure 5A). Neither of these have any significant impact on the target genes tested in our screen. Finally, while the plaque assays are cumbersome such that it is not feasible to text all 61 genes, we have done this for a subset of our “hits” from the siRNA screen, showing that all 5 of those tested show a change in viral progeny that consistent with the screen. These data are now included in Figure 3—figure -supplement 1C.

2) The parallel discussion of the siRNA and AMO data was confusing and more in-depth description of the results is needed. Two of the genes (CLK1 and RAB11FIP3) that showed a significant difference in viral replication and production upon efficient AMO targeting are genes that showed the smallest reduction in viral replication upon siRNA knockdown. Without more information on the siRNAs and the efficiency of the knockdowns, it is difficult to draw conclusions about what the individual genes and splicing events might do. For example, do the CLK1 siRNAs target the exon that was skipped by AMO targeting and therefore both siRNA and AMO treatment gave the same viral phenotype? On the other hand, IQCB1 gave the greatest reduction in viral replication upon siRNA targeting but did not have any effects upon AMO targeting. What does this mean – perhaps that both isoforms are needed for viral replication? The functional consequences of the AS events could only be tested for 11 genes that showed increased exon inclusion on IAV infection. Of these 5 could be modulated (CLK1, RAB11FIP3, PPIP5K2, IQCB1, and REPIN1) – and of these only IQCB1 had an effect greater than 50% upon knockdown. While knockdown of all isoforms may not predict the effect of splice modulation, why not modulate the splicing of genes that showed the most dramatic effect upon knockdown (e.g., ERI2)? In cases where genes showed decreased exon inclusion, could ectopically expressing different isoforms affect viral replication?

We apologize for the lack of clarity in the initial version of our text. We have now clarified the difference in the siRNA experiments (which target all isoforms) and the AMO experiments (which specifically block the viral-induced isoform). As such, these are very different experiments with different goals, which we are more careful now to explain in the text. We have also added data demonstrating the efficiency of knockdown (new Figure 3—figure supplement 1A), as described above. It is true that for some of the genes (i.e. IQCB1) we cannot directly link viral-induced splicing to function; however, it may be that the viral-induced isoform contributes to a stage of viral infection that we have not specifically assayed here.

For reasons we don’t understand, but likely has to do with sequence context, not all alternative exons are able to be blocked by AMOs. This is the case for the 6/11 genes that had induced exon inclusion but for which we were unable to force exon skipping – which we have now clarified in the text. To address the reviewers’ comments and add additional examples of functional impact of splicing on IAV infection, we have now also targeted the alternative 3’ splice site choice in ERI2 and show in new additional data in Figure 4 that this results in a dramatic loss of viral infection and replication.

3) The authors assert that hnRNP K accumulates to higher levels in nuclear speckles during IAV infection by performing immunofluorescence (IFA) (Figure 6). However, differences in hnRNPK localization +/- IAV was not evident in the IFA images provided. The quantification in Figure 6D suggests there are subtle differences, but these are not easily discerned upon visual inspection of Figure 6C. The authors should consider other means to quantify the immunofluorescence, perhaps by measuring the variance in the fluorescence across the nucleus. Another important control here is to perform a similar analysis on SC35 to assess the possibility that IAV infection increases the size of the nuclear speckles (and thus all proteins in them). Are these changes specific to hnRNPK? hnRNPK is highly abundant, so how to determine if a slight increase in its nuclear speckle concentration is functionally relevant? Do small changes in its overall concentration affect splicing? Some of these issues will be difficult to address, but the interpretations of the data should consider them.

We recognize that the changes in hnRNP K localization are not large, although these changes are reproducible and statistically significant. As suggested by the reviewers, we have now measured the volume of nuclear speckles, using SC35 as a marker, with and without infection. We feel these measurements are more precise and accurate than quantifying variance of fluorescence across the nuclei since observed changes at the nucleoplasm cannot be specifically attributed to changes in hnRNP K at nuclear speckles as hnRNP K is localized throughout the nucleus. Additionally, the small nature of nuclear speckles containing hnRNP K compared to hnRNP K present in the entire area of the nucleus may also fail to detect significant changes at nuclear speckles. Based on the SC35 staining, we have found that nuclear speckle volume on average does increases upon infection, at a level comparable to, but somewhat less, than the level of increase of hnRNP K in nuclear speckle upon infection. This suggests that the increase in speckle volume and hnRNP K recruitment to speckles are coordinated with each other. We have added this volume measurement to the figure in a new Figure 6E. We also acknowledge that the increased recruitment of hnRNP K to speckles that we observe upon infection is not large and may not be sufficient to fully account for the changes in splicing. As it is the only significant difference we observe in hnRNP K, we do feel it is important to report and could be driving changes in hnRNP K function, especially in combination with competition between host and viral transcripts; however, we have adjusted the text to be parsimonious in the description of the data and our conclusions.

4) The link between hnRNPK and splicing changes that occur during IAV infection is supported mostly by correlational data (Figure 5); direct evidence that hnRNPK activity is responsible for the IAV-induced splicing changes is lacking. A more direct link might be provided by analyzing a panel of genes that undergo similar isoform shifts during IAV infection and hnRNPK knockdown. Among such genes, does knock down of hnRNPK in IAV infected vs. uninfected cells lead to an additive effect on PSI changes or no further change (suggesting hnRNPK is the primary means by which IAV alters splicing)? Similarly, given that the ∆K-mut IAV appears to be more sensitive to hnRNP K depletion than wt IAV (Figure 4A), perhaps splicing modulation of certain genes would reveal a stronger phenotype in ∆K-mut infection.

We have done the suggested experiment of knockdown of hnRNP K in IAV infected cells and now include this in a new Figure 6—figure supplement 1. In brief, upon KD of hnRNP K, we don’t see any further change in splicing of target genes tested (CLK1 CREB1, etc.) upon IAV infection. While these results are consistent with our model that hnRNP K drives these splicing changes during IAV infection, and we state such now in the text, we are cautious in our interpretation because viral infection is also attenuated upon hnRNP K KD, as shown in Figure 5A. Therefore, it is difficult to fully conclude that the lack of further change in splicing in these cells is only due to the direct role of hnRNP K on these targets and not to reduced infection. This same complication of reduced infection is also true of the ∆K-mut IAV, such that we have not looked at splicing of target genes in this condition since the expectation and interpretation are so unclear.

5) The authors did not cite or discuss several papers that are pertinent to this study that have also profiled IAV-induced host gene expression changes at multiple levels (transcription, splicing, mRNA turnover, translation). Some examples: Bercovich-Kinori et al., 2016, performed RNA-Seq and ribosome profiling in IAV-infected cells, showing a prominent host shutoff phenotype. Bauer et al., 2018, used NETseq to study host transcription during IAV infection. Gaucherand et al., 2019, directly assessed the impact of IAV infection on the accumulation of spliced host transcripts, showing that the flu PA-X protein degrades spliced transcripts more efficiently than unspliced transcripts, and this is mediated through interactions between PA-X and proteins involved in splicing including hnRNPK. Some of these studies used PR8 rather than WSN, but WSN (because of its NS1-mediated targeting of Pol2) should have an even more prominent effect on host mRNA depletion. This previous work provides an essential background for the study presented here and should be discussed.

We apologize for these oversights in our description of the previous literature and have revised the Introduction and Discussion to include discussion of these references, as well as others that have been published since the initial submission.

6) The authors contend that the changes in AS were not due to innate immune mechanisms because these changes were not seen with IFN treatment. The authors should consider that the innate immunity observed early in viral infection (in the case of flu 6-12 h pi) is best modeled by combined IFN treatment and dsRNA activation, not IFN alone. In future studies, it would also be interesting to test influenza virus mutants that are unable to antagonize the IFN response and see what types of splicing changes occur.

We apologize that our comments on this point were unclear in the original manuscript. Our rationale for testing the impact of IFN was to address if the “viral-induced” changes were a secondary response to secreted IFN in bystander cells. We have now clarified the text regarding our rationale and conclusions from this experiment. We agree it will be interesting in future studies to address how much of the host response to viral infection is contributing to splicing changed in virus-containing cells. However, we view this as outside the scope of the current study.